# An Inferred Ancestral CotA Laccase with Improved Expression and Kinetic Efficiency

**DOI:** 10.3390/ijms241310901

**Published:** 2023-06-30

**Authors:** Lei Lei, Lijun Zhao, Yiqia Hou, Chen Yue, Pulin Liu, Yanli Zheng, Wenfang Peng, Jiangke Yang

**Affiliations:** 1School of Life Science and Technology, Wuhan Polytechnic University, Wuhan 430023, China; lei_bc@whpu.edu.cn (L.L.);; 2State Key Laboratory of Biocatalysis and Enzyme Engineering, College of Life Science, Hubei University, Wuhan 430062, China

**Keywords:** ancestral sequence inferring, heat-resistant, laccase, pH stability, dye decolorization

## Abstract

Laccases are widely used in industrial production due to their broad substrate availability and environmentally friendly nature. However, the pursuit of laccases with superior stability and increased heterogeneous expression to meet industry demands appears to be an ongoing challenge. To address this challenge, we resurrected five ancestral sequences of laccase BsCotA and their homologues. All five variants were successfully expressed in soluble and functional forms with improved expression levels in *Escherichia coli*. Among the five variants, three exhibited higher catalytic rates, thermal stabilities, and acidic stabilities. Notably, AncCotA2, the best-performing variant, displayed a *k_cat_*/*K_M_* of 7.5 × 10^5^ M^−1^·s^−1^, 5.2-fold higher than that of the wild-type BsCotA, an improved thermo- and acidic stability, and better dye decolorization ability. This study provides a laccase variant with high application potential and presents a new starting point for future enzyme engineering.

## 1. Introduction

Laccase (EC 1.10.3.2) is a multi-copper containing oxidase that is widely distributed in various organisms such as plants, fungi, bacteria, and some insects. As one of the earliest described enzymes, laccase was first extracted from the sap of lacquer trees in 1883 [1]. It gained recognition for its crucial role in the manufacture of lacquer [2]. Laccase utilizes oxygen as an oxidant to efficiently catalyze the oxidation of phenolic compounds and similar substrates, reducing oxygen into water without releasing radicals [3,4]. Moreover, laccase exhibits high substrate promiscuity and can oxidize a broad range of compounds, including aromatic and non-aromatic compounds, substituted phenols, inorganic ions, and a variety of non-phenolic compounds [3,5]. With its “green” oxidizing property and wide substrate range, laccase has found diverse applications in industries such as pulp industry, textile industry, wastewater treatment, food processing, chemical industry, and herbicide degradation [6,7,8,9].

The large-scale application of laccases presents a promising solution to many industrial challenges. However, their widespread utilization is constrained by several limitations. In industry, the pursuit of enzymes with higher thermostability is an ongoing quest. This is primarily driven by the fact that elevated temperatures are frequently employed to accelerate the reaction rate, while the activity of enzyme products is adversely impacted by drying during production. Consequently, enhancing the activity and stability of enzymes is crucial for developing more robust and potent enzymes for industrial application and experiments.

The laccase BsCotA from *Bacillus subtilis* displays exceptional heat tolerance, with a half-life of inactivation at 80 °C for 2 h [10]. However, its practical application is hindered by folding difficulties that limit protein yield in *Escherichia coli* and *Pichia pastoris* [11,12,13]. These limitations restrict the broad application prospects of BsCotA.

To address this challenge, ancestral sequence resurrection was employed in this study. It is a method that employs various models to construct a phylogenetic tree and infer the amino acid sequences of ancestral nodes from a collection of contemporary, evolutionarily related protein sequences [14]. This method has been widely used to improve thermal stability [14,15,16]. Researchers have reported significant improvements in the thermostability of ancestral proteins using this method, such as the ancestor of the serum paraoxonase (PON) enzyme family, which displayed a 30 °C higher melting temperature than extant human enzymes [17], and the hyper-stable ancestor of β-lactamase, which exhibited a 35 °C higher denaturation temperature compared to the thermophilic *Bacillus licheniformis* [18]. 

In this study, ancestral sequences of BsCotA-like laccases were inferred to obtain variants with improved stability and expression level. The best variant displayed a *T*_50_ value up to 75.70 °C, which is up to 9.23 °C higher than that of BsCotA, and 5.2-fold improved *k_cat_*/*K_M_*. These results demonstrate the potential of ancestral sequence reconstruction to generate protein variants with improved properties and suggest that this method could be applied to other proteins with folding difficulties to broaden their applications.

## 2. Results

### 2.1. Inferring BsCotA Laccase Ancestor

The thermostable laccase BsCotA [19] was used to search against the NCBI (www.ncbi.nlm.nih.gov, accessed on 4 April 2020) protein database to infer ancestral laccase variants. The sequences that showed identities higher than 55% were retrieved (Appendix A), and the redundancy was minimized using CD-hit with 95% cutoff. The sequences were aligned using MUSCLE [20], followed by removing of gaps by trimAl [21]. A maximum likelihood phylogenetic tree was built from this alignment (Appendix A). The CotA sequences were clustered in five major clades and exhibited considerable divergence (Figure 1A). We attempted to reconstruct five ancestral nodes to present the divergence of the CotA-like laccaes. Five ancestral nodes were chosen for ancestral sequence inferring by using the maximum-likelihood algorithm FastML [22]. The ancestral sequence of CotA was named “AncCotA”. AncCotA1 presents the ancestor of clade 1 and clade 2, which has ~90% average identity to the extant BsCotA. AncCotA5 was considered as the last common ancestor of the sequences presented in the phylogenetic tree (Figure 1B). The inferred ancestral sequences showed variations compared to the extant CotA proteins; AncCotA1 had 20 amino acids that differed from BsCotA, while AncCotA5 had 159 residues that differed from BsCotA (Table 1).

Synthetic genes encoding five ancestral constructions and wild-type BsCotA were optimized for *E. coli* codon usage and cloned to pET21a with a C-terminal His-tag. Similar to wild-type BsCotA, all of the five ancestral variants showed a clear single band with an apparent molecular weight of ~60 kDa on SDS-PAGE and displayed two- to fivefold higher expression levels of soluble and functional enzymes compared with the wildtype BsCotA (Appendix A).

### 2.2. Ancestral Inferring Improved Catalytic Efficiency and Stability

The kinetic parameters of the wild-type BsCotA and the inferred variants were investigated using ABTS as the substrate (Table 1 and Figure 2A). Notably, compared with the wildtype BsCotA, AncCotA2 exhibited a *k_cat_*/*K_M_* value of 7.52 × 10^5^ M^−1^·s^−1^, which led to a 5.22-fold improvement compared to the wildtype BsCotA.

As outlined in the introduction, we aimed to improve the stability of heterogeneously expressed BsCotA. To test the response of inferred ancestral variants to heat and pH stress, we incubated the variants at various temperatures and measured the residual activity at 37 °C. In this study, recombinant BsCotA produced in *E. coli* exhibited an apparent *T*_50_ (the temperature at which the residual activity is half of the initial activity) of 66 ± 4 °C (Figure 2B), which was lower than that of the native form isolated from *B. subtilis* endospore coat (75 °C) [10]. The inferred variants, including AncCotA1, AncCotA2, and AncCotA5, exhibited distinctly higher *T*_50_ than the wildtype, with AncCotA2 showing a *T*_50_ of 76 °C, which was up to 9.2 °C higher than that of BsCotA (Table 1).

We further examined the pH stability of the inferred variants under different conditions, representing possible industrial application scenarios. AncCotA4 and AncCotA5 were not tested because of their relatively low activity. All the tested variants and the wild-type exhibited similar stability in alkaline environment, with more than 80% residual activity after incubation in buffer (pH ≥ 8) (Figure 2C). However, after incubation in an acidic buffer, the residual activity of the wildtype BsCotA was approximately 10%, whereas those of AncCotA1, AncCotA2, and AncCotA3 were around 30%, 50%, and 70%, respectively. All five designed variants exhibited similar optimum pH compared to the wildtype BsCotA (pH 4.0) (Appendix A).

### 2.3. AncCotA2 Showed Higher Application Potential

The potential applications of laccases in industry, such as lignin degradation and textile dye decolorization, often require the presence of mediators [23,24]. Among these mediators, ABTS is one of the most commonly used. Therefore, our inferred ancestral variant, which exhibits improved ABTS oxidation ability and stability, has high application potential for various applications. To illustrate the concept, we compared the decolorization efficiency of indigo carmine using AncCotA2 and the wildtype BsCotA.

To investigate the effects of different protein concentrations on indigo carmine decolorization, decolorization reactions were carried out using different laccase concentrations with 0.1 mg/mL indigo carmine and 0.1 mM ABTS as mediator at pH 4. As a control, no decolorization was observed in the absence of ABTS (Appendix A). Both the wildtype BsCotA and AncCotA2 showed rapid initial decolorization rates at all protein concentrations, particularly within the first 10 min of incubation (Figure 3 and Appendix A). After 2 h of incubation, the decolorization rate reached a bottleneck (Appendix A). However, AncCotA2 showed better decolorization performance that the wildtype BsCotA at the same concentration. For example, at a concentration of 3.33 μg/mL, AncCotA2 completely decolorized the indigo carmine after 50 min of incubation, while the wildtype BsCotA required 2 h. At lower concentrations, both AncCotA2 and the wildtype BsCotA failed to completely decolorize and reached a bottleneck. Furthermore, AncCotA2 exhibited a slightly superior final decolorization performance than the wildtype BsCotA; at a concentration of 1 μg/mL, the decolorization rate of AncCotA2 reached 72.1%, whereas the wildtype BsCotA achieved only 65.5% (Figure 3).

### 2.4. Plausible Structural Explanation for the Effect of Residues on the Stability of AncCotA2

To investigate the molecular mechanism underlying the improvement of AncCotA2, the best-performing variant, we used the TrRosetta server to predict its structure [25,26]. The overall structure was highly similar to that of the wildtype enzyme, with a root-mean-square deviation (rmsd) 0.164 Å over Cα atoms (Figure 4A), while the mutated sites of AncCotA2 were distributed throughout the structure (Figure 4B). AncCotA2 had a topology similar to that of BsCotA, which consists of three cupredoxin domains, each comprising seven strands in two beta-sheets arranged in a greek-key beta-barrel.

To enhance the heterologous expression and stability of proteins, several common rational design strategies can be employed. These strategies often involve the introduction of hydrogen bonds [27], optimization of core packing [28], and enhancement of surface polarity and charge distribution [29]. Here, in this study, we analyzed the impact of all 69 sites that differed from BsCotA and identified some amino acid substitutions that might enhance stability, as outlined in Appendix A. Seven of the 69 mutated sites may be related to improved core packing. For instance, the substitution of threonine to methionine at site 196, located within the hydrophobic core of the protein, improves the core packing by introducing a non-polar side chain of methionine, thereby increasing the protein folding efficiency. Additionally, hydrogen bonds are a key determinant of protein stability, and three sites were found to form a new hydrogen bond. Following the replacement of isoleucine to threonine at site 438, the introduced hydroxy group formed a hydrogen bond with the mainchain carboxyl group of Proline417. The introduced hydrogen bond may improve the thermal and acidic stability of BsCotA2. Thirteen sites were found to have improved surface polarity/charge. For example, the substitution of hydrophobic residues with hydrophilic residues such as A439E increase the surface charging, thus this might improve the protein expression level.

We also observed 25 sites with conservative replacement, which may not alter the properties of the variants, and 21 sites that were challenging to explain the roles in our ancestral variants. Notably, the residues at the active site showed a high degree of alignment (Figure 4A). Consequently, despite those 69 substitutions in AncCotA2 and it having 5.22-fold higher *k_cat_*/*K_M_*, 5.88-fold higher expression in *E. coli*, and 9.23 °C higher heat tolerance compared with the wildtype BsCotA, the active site of AncCotA2 remained similar to that of BsCotA.

## 3. Discussion

Laccase is an oxidoreductase enzyme that participates in the catalysis of a diverse array of substrates, including lignin, aromatic compounds, and phenols. The reported specific activities seem to range from ~20 units (μM ABTS min^−1^ mg^−1^ protein) for the native form of *Trametes orientalis* laccase to ~400 unit for the OB-1 mutant of *Basidiomycete* PM1 laccase (Table 2) [8,30]. BsCotA, a laccase isolated from *B. subtilis* endospore, has good thermostability [10,19]. However, most of the recombinantly expressed CotA proteins and their orthologs from other *Bacillus* strains were found to aggregate in the inclusion bodies and displayed decreased stability when expressed in *E. coli* [31,32,33]. As a result, the loss of thermostability and expression difficulty during heterologous expression limit the further application of BsCotA. To overcome these limitations, a rational design of CotA from *Bacillus* sp. HR03, E188K, exhibited threefold higher thermal activation and better tolerance to organic solvents [34,35].

In this study, we inferred five different ancestral BsCotA variants and characterized their thermal stability, pH stability, and enzymatic activity. All five variants exhibited different levels of improved expression, and the AncCotA2 variant exhibited the most significant improvement, with a 9.23 °C higher *T*_50_ value, improved acidic stability, and ~5.2-fold improved *k_cat_*/*K_M_*. The expression level of the wild-type BsCotA in this study was quite low. In contrast, Durão et al. produced BsCotA in *E. coli* strain Tuner (DE3), purified it using a cationic exchange SP-column and gel filtration, and obtained up to 23 mg of the wild-type BsCotA from 1 L culture [37]. While in our study, the BsCotA and its variants were produced in *E. coli* strain BL21 (DE3), purified using Ni-NTA beads. The observed difference in results might be attributed to variations in the *E. coli* host/plasmid and purification methods used. Nonetheless, the expression level improvements in our study were based on the higher folding efficiency of the inferred ancestral sequence, which is expected to be applicable to any given heterogeneous expression system. A comparison between *Basidiomycete* laccase and the ancestral variants revealed that two of the three variants showed higher expression in *Saccharomyces cerevisiae*, improved thermal and pH stability, and similar *k_cat_*/*K_M_* with those of the wildtype [38]. Based on the inferred ancestral sequence, the expression level of a laccase from the medical mushroom *Agaricus brasiliensis* increased from 18 U/mL to 2700 U/mL, and it showed greater robustness to temperature and pH [39]. These findings, along with the results of our study, suggest that the enhanced stability and expression levels observed with the inferred ancestors are not limited to specific expression hosts.

The thermal stability of our laccases warrants discussion. In our experiments, we have observed significant differences between the wild-type BsCotA and the native form reported in the literature [10]. We hypothesize that the lack of appropriate molecular chaperones during the heterologous expression in *Escherichia coli* may have resulted in improper folding or folding into a non-native conformation, leading to decreased stability. However, despite these circumstances, all proteins were expressed, purified, and characterized under consistent conditions. Therefore, we can confidently conclude that AncCotA2 exhibits superior stability compared to the wild type BsCotA.

The method of inferring ancestral sequence has been widely used to improve enzyme stability and expression in vitro, as reported in various studies [18,40,41,42], which have shown improved *k_cat_* and/or *k_cat_*/*K_M_* [17,18,43]. For instance, a previous study on LeuB, a 3-isopropylmalate dehydrogenase, reconstructed an ancestral sequence that exhibited a fourfold higher *k_cat_*/*K_M_*, compared to the extant enzyme [44]. Similarly, a study on fungal laccase inferred three ancestral sequences, two of which were expressed successfully and exhibited improved expression levels, better resistance to heat and acidic pH, and similar *k_cat_*_/_*K_M_* compared with modern laccases [38]. The best-performing variant LacAnc100 displayed 2–5-fold improved *k_cat_* for all the tested substrates, which is consistent with our present work and prompted us to investigate the mechanism underlying the improvement in catalytic performance. Our modeled structure analysis revealed no significant changes in the substrate-binding region between ancestral sequence and the wildtype BsCotA (Figure 4), suggesting that the improvement could be attributed to additional factors. Based on the pH stability results, we observed that BsCotA lost its activity completely after 30 min of incubation under the optimum pH for ABTS oxidation (pH 4) (Figure 2C) whereas AncCotA2 retained approximately 70% of its activity under the same condition. The acidic stability of AncCotA2 determines its ability to maintain its structure and functionality under pH 4. The wild-type BsCotA is unstable and undergoes denaturation or structural changes under the same condition. Thus, we speculated that the improved *k_cat_*/*K_M_* observed in the ancestral sequence could be attributed to its improved robustness to the reaction condition, which could lead to the inactivation of the wild-type BsCotA.

The results of indigo carmine decolorization are noteworthy. The reaction pH was set to pH 4, which is the optimal condition for ABTS oxidation. In both variants, no decolorization was detected in the absence of ABTS. This finding is consistent with previous studies [33], which have shown that CotA laccase cannot directly decolorize indigo carmine but the presence of the mediator ABTS could significantly enhance the decolorization [45,46]. When 3.33 μg/mL protein was used, the decolorization of both the wildtype BsCotA and AncCotA2 exhibited linear responses (see Figure 3 and Appendix A). Because the input ABTS concentration (0.1 mM) is below the *K_M_* (0.41 mM for AncCotA2 and 0.88 mM for the wildtype BsCotA), both variants were unable to reach their maximum catalytic rates, resulting in similar initial decolorization rates. After 10 min incubation, the decolorization rates of both variants entered a slowly rising region. The wildtype BsCotA reaction proceeded very slowly and reached complete decolorization after 1.5 h. Whereas, the AncCotA2 reaction exhibited less inhibition and reached complete decolorization in 40 min. These results suggest that the improved decolorization observed in AncCotA2 is a result of its enhanced stability in acidic buffer environment, as discussed above. The experiments comparing AncCotA2 and the wildtype BsCotA in indigo carmine decolorization demonstrated that AncCotA2 outperformed the wildtype BsCotA in terms of both decolorization speed and final decolorization rate, particularly at low protein concentrations. In summary, the inferred ancestral variant of laccases, AncCotA2, showed improved ABTS oxidation ability and stability, which makes it a promising candidate for industrial applications such as textile dye decolorization.

In general, improved stability is associated with a faster refolding rate [47,48]. Moreover, there is a correlation between heterogenous expression and refolding rate, where only one or two residues with back-to-ancestor replacement can have significant impact [49]. These findings suggest that the effect on refolding rate may contribute to the improvement of protein stability and expression level in ancestral sequence resurrection. This leads to a fundamental question: what are the factors that contribute to the improvement during ancestral sequence inferring? We attempted to provide some explanations for several sites from the predicted protein structure (Figure 4), such as the introduced hydrogen bonds and hydrophobic core packing that could increase the thermostability and folding efficiency [28]. However, this attempt alone cannot explain all of the substitution sites. It is possible that the improved stability and expression level are due to the unpredictable combinations of the inferred mutations, which alter the dynamic properties of the protein, rather than the simple superposition of the effect of individual sites.

“Assuming” that the inferred ancestor is the same as the actual ancestor, the evolutionary reasons behind improved stability and expression level in ancestral sequences are intriguing. First, the ancestral inferring of the “corrected” rare residues that differed from the majority of the extant family members at each site might increase the stability and folding efficiency [17,50,51]. Second, as is well-known, proper chaperones play an important role in assisting protein folding [52]. In the case of ancestral proteins, it is possible that they were expressed in the absence of proper chaperones. This absence may have served as an evolutionary pressure, driving proteins towards higher folding efficiency and stability [17,33,53]. This hypothesis provides an explanation for the improved heterogeneous expression and stability of ancestral proteins.

In the case of this study and probably in other studies, it will be difficult to exclude the existence of bias during the ancestral sequence inference. Thus, the improvement of stability could be an artifact due to this bias [17]. In the future, new algorithms that reduce this bias, such as ProtASR2 [54], may bring a better understanding of the correlation between improved stability and ancestral sequence.

In conclusion, we inferred and developed five possible ancestral variants to improve the stability of BsCotA during recombinant expression. AncCotA2 obtained the improved stability and also the best kinetic efficiency, with 5.2-fold improvement of *k_cat_*/*K_M_* compared with the wildtype BsCotA. AncCotA2 also exhibited a higher application potential in dye decolorization. It could be used for further directed evolution or rational design to improve catalytic rates. These results also indicate that the method of ancestral sequence resurrection could be used for the enzyme engineering to fit industrial exploration for better stability and higher expression level.

## 4. Materials and Methods

### 4.1. Ancestral Sequence Reconstruction

The sequences of BsCotA family members were collected from the National Center for Biotechnology Information non-redundant protein sequence database (nr) by using protein alignment BLAST (blastp) tool with its default settings. Sequences with ≥55% identity and ≥95% coverage were included. Redundant sequences with ≥90% identity to other sequences were removed using CD-HIT (version 4.8.1), and alignment was performed with Muscle [20]. The gap positions in the alignment were removed using trimAI (version 1.2) [21]. IQ-TREE (version 1.6.12) was used to test the evolutionary models and reconstruct a maximum likelihood tree. The best-fitting model tested by IQ-TREE was LG + I + G4 [55]. The reliability for the internal nodes was assessed by bootstrapping (10,000 bootstrap replicates). The most likely ancestral sequence was inferred using the FastML (version 3.11) algorithm that exhibited high posterior probability [16,22,56], based on the alignment and phylogenetic tree generated by IQ-tree. The substitution model utilized was LG, along with a gamma distribution.

### 4.2. Protein Expression and Purification

BsCotA (Uniprot number: P07788) and designed variants were codon-optimized for expression in *E. coli* by DNAworks [57]. BsCotA and variants were synthesized (Genscript, Nanjing, China). The synthetic genes were cloned into a pET21a vector (Novagen, Darmstadt, Germany) in fusion with a C-terminal His-tag by using EcoRI and SalI restriction sites. The constructed plasmids were transformed into *E. coli* BL21 (DE3) cells with chaperone plasmid pGro7 (Merck, Darmstadt, Germany) for expression.

BsCotA and its variants were produced as described [37]. Cells containing the recombinant plasmids were grown overnight in 5 mL of LB medium at 37 °C. The cultures (250 mL LB cultures) were grown at 37 °C until OD_600_ reached 0.5–0.6, moved to 25 °C, and supplemented with IPTG in 0.1 mM and 0.25 mM copper sulfate. After 4 h of incubation, the shaker was switched off to achieve a microaerobic condition. After further culture for 24 h, cells were harvested by centrifugation at 4 °C, resuspended in 50 mL of lysis buffer (50 mM Tris, 100 mM NaCl, 1 mM CuSO_4_ (pH 8.0), and 50 U benzonase), and homogenized using a high-pressure cell disrupter (JNBIO, Inc., Guangzhou, China). The supernatant was collected after centrifugation. The lysates were clarified by centrifugation and loaded on an IMAC (Immobilized Metal Affinity Chromatography) column (Bio-Rad, Hercules, CA, USA) equipped with a protein purification system (JP-blupadStar, Shanghai, China). The columns were washed with 50 mL of lysis buffer, followed by 100 mL of lysis buffer with 35 mM imidazole. The bound proteins were eluted by lysis buffer supplemented with 250 mM imidazole. Fractions containing the expressed proteins were analyzed by SDS-PAGE and enzyme activity assay. The purified protein was concentrated by ultrafiltration (Amicon Ultra, Sigma-Aldrich, St. Louis, MO, USA). Imidazole was removed by dialysis in dialysis buffer (50 mM Tris, 100 mM NaCl, 1 mM CuSO_4_, pH = 8.0). The purified proteins were concentrated to ~2 mg/mL and stored at −80 °C. Final enzyme concentrations were determined by the BCA assay [58].

### 4.3. Enzyme Assay

Laccase activity assays were routinely carried out at 37 °C with 5 mM ABTS in Britton–Robinson buffer (mixture H_3_PO_4_-HAc-H_3_BO_3_, pH = 4). The reaction mixture was incubated at 37 °C for 5 min and quenched by adding an equal volume of methanol. The oxidation of ABTS was monitored using a spectrophotometer at 420 nm (ε = 36,000 M^−1^ cm^−1^) [59]. Laccase activity was evaluated as a function of pH in Britton–Robinson buffer (pH range 3–12) for ABTS (5 mM).

The optimal pH was determined in Britton–Robinson buffer with different pH values (3.0–12.0) and 1 mM ABTS and 3.33 µg/mL enzyme. Kinetic parameters of the purified laccase were determined at 37 °C against different concentrations of ABTS (0 μM–2 mM, pH = 4). Wild-type BsCotA and its variants (3.33 µg/mL) were reacted with ABTS for 5 min. In these conditions, the rates of ABTS oxidation were found to be linear up to 5 min. Enzymatic assays were performed in triplicate. One unit was defined as the amount of enzyme that oxidized 1 μmol substrate per minute.

### 4.4. Thermostability and pH Stability

Aliquots of BsCotA variants (20 µL of 1 µM protein) were incubated at a range of temperature for 30 min. The samples were diluted to 300 µL in an activity buffer with 5 mM ABTS as the substrate to measure ABTS oxidation activity. The temperature at which the BsCotA variants lost 50% of its initial activity after 30 min of incubation was denoted as *T*_50_. Kinetic thermostability and *T*_50_ were calculated by fitting its residual activity at different temperatures to a 4-parameter Boltzmann sigmoidal curve as described in [60]: $$A_{t}=A_{0}+\frac{A_{f}-A_{0}}{1+e^{\left(T_{50}-T\right)/m}}$$
where *A_t_* corresponds to the residual activity following incubation at a given temperature *T*, *A*_0_ is the activity of the untreated control incubated on ice, *A_f_* is the activity at maximal inactivation temperature, and *m* is the sigmoidal slope coefficient. To display, the normalized rates were fitted to a 4-parameter Boltzmann sigmoid using the GraphPad Prism 7.0 (GraphPad Software, San Diego, CA, USA). All measurements were conducted in triplicate.

pH stability was determined by incubating 3.33 µg/mL enzyme in Britton–Robinson buffer with different pH levels (3.0–12.0) at 37 °C for 30 min. Residual enzyme activity was determined at 37 °C and under the optimum pH of 4.0. The enzyme activity of the untreated enzyme was taken as 100% to evaluate the wild-type and mutant laccases.

### 4.5. Decolorization of Indigo Carmine (IC)

In brief, 0.1 mg/mL indigo carmine was dissolved in Britton–Robinson buffer (pH 4.0) and incubated with 0.1 mM ABTS as the mediator to evaluate the dye decolorization ability of the laccase variants. The reaction was initiated by the addition of laccase, which was substituted with water as the control. The reactions were performed in 96-well plates at 30 °C with shaking for a period of 12 h at intervals of 10 min. Absorbance at 602 nm was recorded to follow IC decolorization. The decolorization rate of IC by the tested laccase variants was calculated.
Decolorization (%) = (1 − ((*A* − *A*_0_)/*A*_100_)) × 100%
where *A* is the absorbance of indigo decolorized by ancestral laccase when ABTS is the mediator, *A*_0_ is the initial absorbance of indigo dye and ABTS, and *A*_100_ is the absorbance of indigo.

### 4.6. Structural Modeling

Three-dimensional models of inferred variants were generated using the TrRosetta server (https://yanglab.nankai.edu.cn/trRosetta/, accessed on 3 May 2021), with default parameters and with restraint from both deep learning and homologue templates [25]. All the structural figures were prepared using PyMOL (Schrödinger).

## Figures and Tables

**Figure 1 ijms-24-10901-f001:**
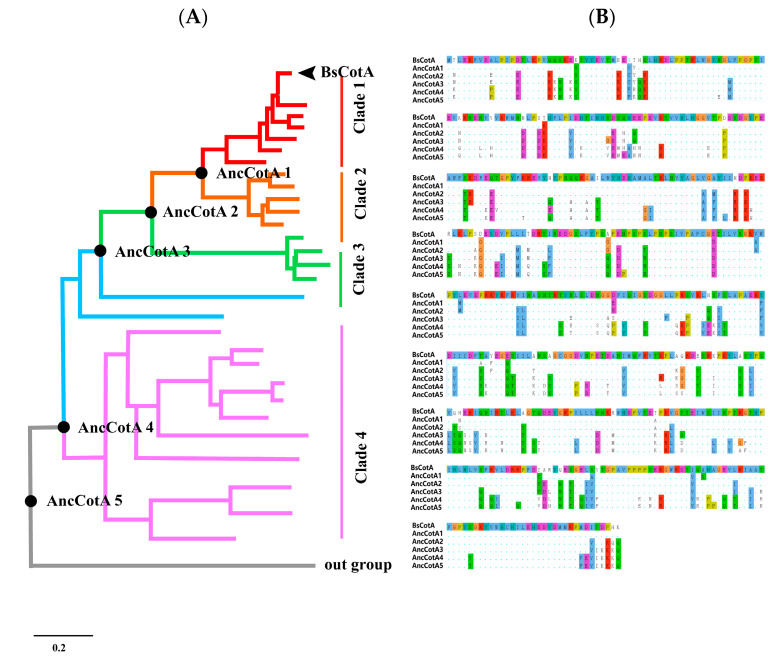
(**A**) Phylogenetic tree of BsCotA. Lineages are colored according to sequence similarity (from top to bottom, 90–100%, 70–80%, 60–70%, and 55–60%). The ancestor sequences are marked with black dots (from top to bottom, AncCotA1, 2, 3, 4, and 5). The black arrow refers to BsCotA. (**B**) Alignment of the five BsCotA variants and the inferred CotA variants. The BsCotA sequence is shown in the head, the residues that differ from BsCotA are highlighted, and the consensus sequence is shown as dots, the amino acids are coloured in clustal colour scheme.

**Figure 2 ijms-24-10901-f002:**
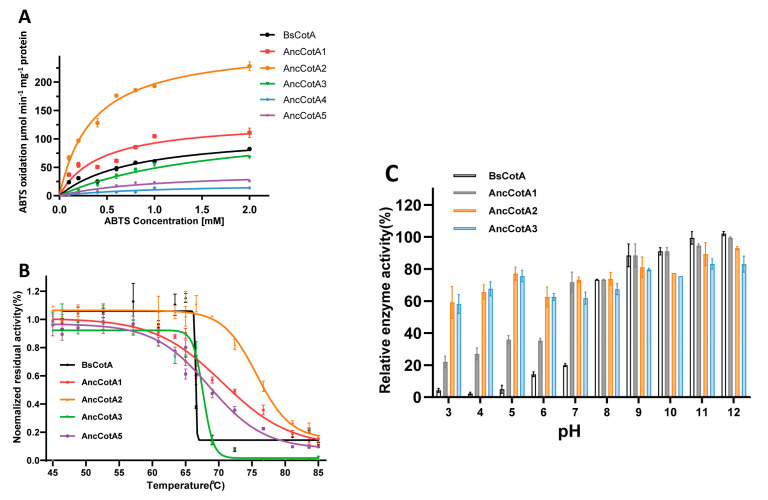
Characterization of inferred BsCotA ancestral variants. (**A**) Kinetic parameters of five ancestral variants and wild-type BsCotA. Data points represent the average specific activity of three independent measurements, and the error bars represent the standard deviation. The lines are the direct fit to the Michaelis–Menten equation (*R*^2^ ≥ 0.97). (**B**) Resistance of BsCotA variants to thermal denaturation. The studied BsCotA variants were incubated in pH 4 at different temperatures for 30 min. Residual enzymatic activity was then measured at 37 °C. Relative residual activity was derived from the initial rates of ABTS oxidation and plotted as a fraction of the activity of the enzyme incubated on ice. The dots represent the mean of residual activity, and the error bars represent standard deviation. The fit is based on a four-parameter sigmoidal curve (Materials and Methods). Measurements were done in triplicate and values normalized to the enzyme incubated at 25 °C. (**C**) pH stability of wildtype, AncCotA1, AncCotA2, and AncCotA3 (Materials and Methods). BsCotA and its variants were incubated in Britton–Robinson buffer with different pH values. After incubation for 30 min, residual activities were measured at 37 °C with 2 mM ABTS in Britton–Robinson buffer (pH 4.0). The columns represent residual activity, and the error bars represent standard deviation. Measurements were done in triplicate and values normalized to the enzyme incubated at pH 8.

**Figure 3 ijms-24-10901-f003:**
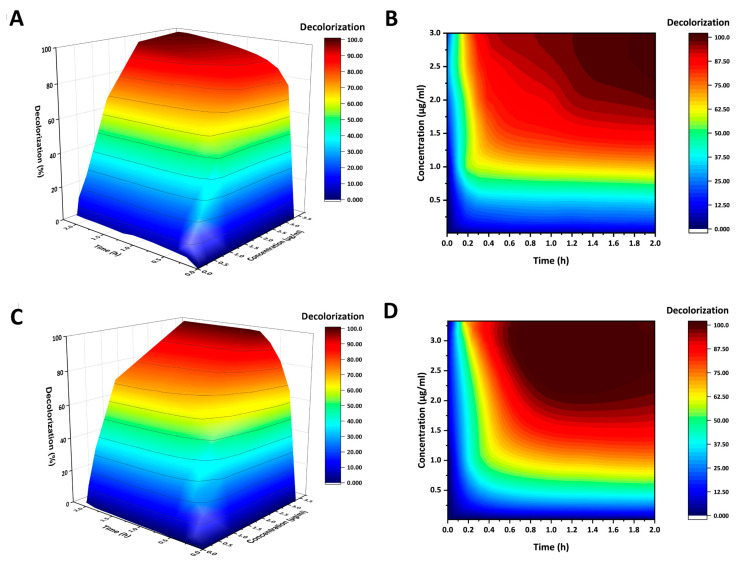
Indigo carmine decolorization within 2 h of incubation with BsCotA and mutant (AncCotA2) shown with 3D surfaces and contour plots. (**A**) 3D surface plot of indigo decolorization rate of BsCotA. (**B**) Contour plot of indigo decolorization rate of BsCotA. (**C**) 3D surface plot of indigo decolorization rate of AncCotA2. (**D**) Contour plot of indigo decolorization rate of AncCotA2.

**Figure 4 ijms-24-10901-f004:**
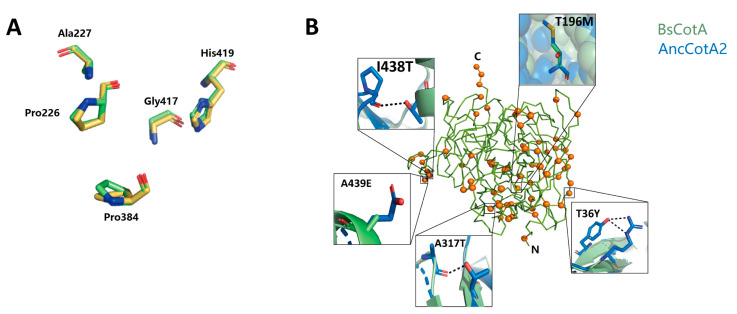
(**A**) Structural alignment of key residues in the vicinity of the catalytic triad in the modeled structure of AncCotA2 (coloured in yellow) compared with BsCotA (PDB: 2X87, coloured in green) without Cu and ABTS. (**B**) Structural details of the inferred variant AncCotA2. Wild-type BsCotA is shown in green, and the inferred positions are distributed throughout AncCotA2 and highlighted as orange spheres. Thumbnails highlight the deduced mutations that contributed to stabilization.

**Table 1 ijms-24-10901-t001:** Stability and kinetic parameters of BsCotA and inferred variants.

CotA Variants	Mut ^a^	Normalized Activity (μmol min^−1^ mg Enzyme^−1^)	Yield (mg/L)	Thermostability (*T*_50_)	ABTS Oxidation		
					*K_M_* (mM)	*k_cat_* (S^−1^)	*k_cat_*/*K_M_* (M^−1^·s^−1^)
BsCotA	0	82.54	0.801	66 ± 4 °C	0.9 ± 0.2	127 ± 15	1.4 × 10^5^
AncCotA1	19	107.48	3.133	71 ± 0.1 °C	0.5 ± 0.1	155 ± 28	3.0 × 10^5^
AncCotA2	69	198.43	4.705	76 ± 0.1 °C	0.4 ± 0.04	306 ± 12	7.5 × 10^5^
AncCotA3	101	68.67	3.083	64 ± 0.3 °C	1.9 ± 0.5	152 ± 23	0.8 × 10^5^
AncCotA4	152	13.65	2.317	NA	1 ± 0.4	21 ± 4	0.2 × 10^5^
AncCotA5	159	25.12	2.932	69 ± 0.10 °C	1 ± 0.4	48 ± 9	0.4 × 10^5^

^a^ Number of amino acid mutations relative to BsCotA.

**Table 2 ijms-24-10901-t002:** Comparison of laccase characteristics with other reported laccases.

Name	Expression Host	Specific Activity (U/mg)	Optimal Temperature	Optimal pH	*K_M_* (mM)	*k_cat_* (s^−1^)	*k_cat_*/*K_M_* (M^−1^·s^−1^)	Ref.
*Trametes trogii*	native form	352.1	45 °C	3	0.069	7985	1.15 × 10^8^	[36]
*Bacillus subtilis*	native form	82.54	75 °C	3	0.106	16.8 ± 0.8	1.52 × 10^5^	[10]
*Basidiomycete* PM1 laccase OB-1 mutant	*S. cerevisiae*	400	60 °C	4	0.0044	110 ± 3	2.47 × 10^8^	[30]
*Bacillus pumilus* CotA	*E*. *coli*	117	70 °C	4	0.080	291 ± 2.7	3.64 × 10^6^	[33]
BsCotA WT	*E*. *coli*	82.54	70 °C	4	0.79 ± 0.23	120.7	1.44 × 10^5^	This study
BsCotA AncCotA2	*E*. *coli*	198.43	70 °C	4	0.35 ± 0.03	288.3	7.52 × 10^5^	This study
*Trametes orientalis*	native form	20.667	80 °C	4	0.33	21.81	ND	[8]
*Trametes hirsuta.*	native form	22.111	50 °C	6	0.087	1.479	1.48 × 10^6^	[7]
*Ganoderma australe* Galacc-F	native form	22.214	55 °C	6	0.164	ND	1.66 × 10^6^	[9]

## Data Availability

This material is available free of charge via MDPI website.

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
