# Peer review of "An Inferred Ancestral CotA Laccase with Improved Expression and Kinetic Efficiency"

_ijms, 2023, doi:10.3390/ijms241310901_

Round 1

Reviewer 1 Report

The manuscript should be polished for english language.

Reviewer 2 Report

 This paper discusses the study on laccases, a type of enzyme widely used in industry due to their versatility and environmentally friendly properties. However, the demand for more stable and abundantly expressed laccases has been a challenge. To overcome this, the study introduced five ancestral sequences of a particular laccase called BsCotA, and its homologues. All five variants were successfully expressed in Escherichia coli in soluble and functional forms. The most promising variant, AncCotA2, demonstrated significantly superior characteristics than the wild-type, including improved catalytic efficiency, thermal and acidic stability, and enhanced dye decolorization capability. This research offers a potential high-performing laccase variant and lays the foundation for future enzyme engineering studies.

Appreciate the clear layout of methods 

Major Points

1. The authors noticed a disparity between their results and the previous study by Durao et al. While the authors attributed these differences to variations in E. coli host/plasmid and purification methods, a more rigorous comparison between the conditions could better support this claim.

2. The authors argue that their improved enzyme characteristics should extend beyond the specific expression system they used (E. coli), but did not experimentally test this assumption. Direct testing of their laccase variants in different hosts would strengthen their claims.

3.  The differences in thermal stability observed between their version of wild-type BsCotA and literature reports warrant further investigation. The authors hypothesize that this discrepancy may be due to improper protein folding during heterologous expression, but additional experiments could confirm this hypothesis.

4. The authors find that AncCotA2 outperforms the wild-type enzyme in terms of decolorization speed and final rate, especially at low protein concentrations. However, the correlation between decolorization performance and other improved enzyme characteristics (e.g., thermal stability, catalytic efficiency) could be more thoroughly investigated.

Overall this is an interesting paper that adds further value to field and supports development of improved laccases.

Clear presentation and easy to comprehend
